# *Biomphalaria glabrata* Granulin Increases Resistance to *Schistosoma mansoni* Infection in Several *Biomphalaria* Species and Induces the Production of Reactive Oxygen Species by Haemocytes

**DOI:** 10.3390/genes11010038

**Published:** 2019-12-28

**Authors:** Jacob R. Hambrook, Abdullah A. Gharamah, Emmanuel A. Pila, Solomon Hussein, Patrick C. Hanington

**Affiliations:** 1School of Public Health, University of Alberta, Edmonton, AB T6G 2R3, Canada; hambrook@ualberta.ca (J.R.H.); gharamah@ualberta.ca (A.A.G.); emmanuel.pila@gov.ab.ca (E.A.P.); 2Department of Biological Sciences, University of Alberta, Edmonton, AB T6G 2R3, Canada; skhussei@ualberta.ca

**Keywords:** granulin, *Biomphalaria glabrata*, Schistosoma mansoni, host-parasite compatibility, invertebrate immunology

## Abstract

Gastropod molluscs, which have co-evolved with parasitic digenean trematodes for millions of years, utilize circulating heamocytes as the primary method of containing and killing these invading parasites. In order to do so, they must generate suitable amounts of haemocytes that are properly armed to kill parasitic worms. One method by which they generate the haemocytes required to initiate the appropriate cell mediated immune response is via the production and post-translational processing of granulins. Granulins are an evolutionarily conserved family of growth factors present in the majority of eukaryotic life forms. In their pro-granulin form, they can elicit cellular replication and differentiation. The pro-granulins can be further processed by elastase to generate smaller granulin fragments that have been shown to functionally differ from the pro-granulin precursor. In this study, we demonstrate that in vivo addition of *Biomphalaria glabrata* pro-granulin (BgGRN) can reduce *Schistosoma mansoni* infection success in numerous *Biomphalaria* sp. when challenged with different *S. mansoni* strains. We also demonstrate that cleavage of BgGRN into individual granulin subunits by elastase results in the stimulation of haemocytes to produce reactive oxygen species.

## 1. Introduction

Schistosomiasis is a serious parasitic disease caused by trematodes of the genus *Schistosoma*. It affects over 206 million people worldwide, especially in tropical and sub-tropical regions [1,2]. After malaria and intestinal helminthiasis, schistosomiasis is the third most devastating tropical disease in the world, representing a major source of morbidity and mortality for developing countries in Africa, South America, the Caribbean, the Middle East, and Asia [1].

Freshwater snails of the genus *Biomphalaria* act as obligate intermediate hosts for *Schistosoma mansoni*, which is the predominant causative agent of intestinal schistosomiasis in Africa and South America. This association with human schistosomiasis has made *Biomphalaria* sp. some of the most extensively studied gastropods in terms of immunobiology and host–parasite interactions. Although numerous species are capable of transmitting *S. mansoni*, *B. glabrata* has emerged as the predominant model for studying the intra-molluscan aspects of the *S. mansoni* lifecycle, with hopes that this research will prove fruitful towards disease control efforts [3]. While *B. glabrata* is a strong model, it does not transmit as much *S. mansoni* as its African counterparts. Thus, studying other species of *Biomphalaria* is important to understand whether information gleaned from *B. glabrata* studies translates across all *Biomphalaria* snails. Seeing as the majority of *S. mansoni* transmission worldwide occurs through *Biomphalaria sudanica* and *Biomphalaria pfefferi*, these two snail species are essential to include in such comparisons [4]. In addition, there are different strains of *B. glabrata* that display differing compatibility phenotypes to *S. mansoni* infection. Some strains display high levels of resistance such as BS-90, 13-16-R1, 10-R2 [5,6] while some show high levels of susceptibility such as the M-line and NMRI strains [7,8]. Understanding the molecular differences between these strains and species allows for a more thorough understanding of the molecular underpinnings of resistance.

A wide array of proteins have been identified and studied in *B. glabrata* with the intent to discover those factors that elicit and coordinate the immune response against *S. mansoni* [9,10]. One such protein is *B. glabrata* granulin (BgGRN). Granulins are evolutionarily conserved growth factors that are found in diverse phyla ranging from eubacteria to humans [11]. They are potent growth factors that drive proliferation of immune cells in organisms, spanning the animal kingdom. It has been shown to be essential for several molecular processes in humans and other animals, such as: in tissue repair, cellular proliferation, development, wound healing, and inflammation [12,13,14].

Progranulin proteins (PGRNs) are secreted pro-peptides that can be subsequently cleaved into smaller functional units by elastase [15]. Often, granulin pro-peptides appear to resemble a “pearl necklace” wherein folded granulin domains, held together by disulfide bonds, are strung together while separated by predicted cut sites for elastase [16,17,18,19,20]. The granulin domains flanked by the elastase sites are defined by a characteristic 12-cysteine repeat. *B. glabrata* granulin is comprised of 4 such domains with predicted elastase cut sites before and after each of these 12 cysteine domains, a fact noted by Pila et al. in their earlier work on BgGRN (Figure 1) [11,12,16].

Previous work has demonstrated that full-length BgGRN induces proliferation of *B. glabrata* hemocytes, and that these haemocytes are largely adherent and express Bg Toll-Like Receptor (TLR). Both of these characteristics render haemocytes better capable of killing invading sporocysts [12,21]. Additionally, susceptible *B. glabrata* snails can be made resistant to infection with *S. mansoni* with an injection of full-length rBgGRN, while siRNA mediated knock-down of BgGRN results in a loss of resistance [12]. The differences in resistance profiles seen between M-line and BS-90 *B. glabrata* seems to be somewhat reliant upon the capacity of BS-90 snails to increase expression of BgGRN at an earlier stage of infection. Moreover, analysis of BS-90 plasma reveals the presence of several proteolytically generated BgGRN fragments during initial infection with *S. mansoni.* Whereas M-line plasma features some intermediate forms at ~20 kda, the ~10 kda forms, which likely correspond to single granulin subunits, are only observed in BS-90 plasma following *S. mansoni* challenge [12]. Previous work suggests that granulin fragments are functionally distinct from the pro-granulin protein. For example, individual granulin domains have been shown to be pro-inflammatory and induce the production of IL-8 in mammals [16]. Thus, we sought to expand upon these previous findings and examine whether or not the intermediate and single subunit BgGRNs observed in BS-90 plasma had any function in cell-based immunity, specifically the production of reactive oxygen species (ROS) production which is known to underpin haemocyte-mediated killing of *S. mansoni* sporocysts, and can therefore serve as a marker of immune cell activation [2,22,23,24].

Additionally, we demonstrate that the ability of BgGRN to confer resistance to *S. mansoni* infection is consistent in other *Biomphalaria* species/strains. That BgGRN confers increased resilience to *S. mansoni* infection beyond the *B. glabrata* model suggests that the newly generated haemocytes driven to expand following rBgGRN treatment are broadly important to the anti-*S. manoni* immune response. This marks an important first step in moving beyond the *B. glabrata* model towards translating model system discoveries related to snail-schistosome compatibility into snail species that transmit the bulk of *S. manoni* to humans—a necessity if snail-schistosome research is ever to have a significant impact on disease transmission.

## 2. Materials and Methods

### 2.1. Animal Ethics Statement

*S. mansoni* was obtained from infected Swiss–Webster mice provided by the National Institutes of Health (NIH)/National Institute of Allergy and Infectious Disease (NIAID) Schistosomiasis Resource Center at the Biomedical Research Institute [25]. All animal work observed ethical requirements and was approved by the Canadian Council of Animal Care and Use Committee (Biosciences) for the University of Alberta (AUP00000057).

### 2.2. Snails and Parasites

*Biomphalaria pfeifferi*, *sudanica* and *glabrata* (BS-90, M-line, NMRI and BB-02) snails were maintained in aerated artificial spring water at 23–25 °C, following a 12-h day/night cycle and were fed red-leaf lettuce as needed. All snail exposures were performed using either the NMRI or PR-1 strains of *S. mansoni*.

### 2.3. Recombinant BgGRN Synthesis and Purification

Recombinant full-length BgGRN (rBgGRN) and four of its 2–3 granulin domain-containing intermediates were generated using the Gateway cloning system according to the manufacturer’s instructions (Thermo Fisher Scientific, Mississauga, ON, Canada). Briefly, the granulin coding region (ADX33287.1) was amplified with Phusion high-fidelity DNA polymerase and the specific primers listen in Table 1 before being cloned into the pENTR/D-TOPO vector. Plasmid DNA from this entry clone was isolated and cloned into pIB/V5-His-DEST vector in a Clonase II recombination reaction to produce the expression clone. The identities of BgGRN and its specific domains were sequence-verified after cloning in the entry and expression vectors. Plasmid DNA was extracted from the expression vector and transfected into a Sf9 insect cell line using a Cellfectin II reagent (Thermo Fisher Scientific). Western blot using primary antibodies against the V5 and histidine tags on the recombinant protein confirmed ex- pression of full-length BgGRN and its intermediate domains.

Sf9 cells stably expressing BgGRN and its intermediates were selected and maintained in a medium containing blasticidin. Cultures were scaled up in 75 cm^2^ flasks, and the medium containing secreted BgGRN was filtered through 0.45-μm filters before being purified using HisTrap FF column (GE Healthcare, Mississauga, ON, Canada) according to the manufacturer’s instructions. Purified BgGRN was dialyzed against PBS buffer twice for 2 h each and then once overnight using Slide-A-Lyzer dialysis kit (Thermo Fisher Scientific, Mississauga, ON, Canada).

### 2.4. Anti-BgGRN Polyclonal Antibody Generation and Validation

A mouse anti-BgGRN polyclonal antibody was generated against the recombinant BgGRN (GenScript). The antibody was affinity-purified using a Protein G affinity column (No. 17–0404-01; GE Healthcare), and then further purified against recombinant BgGRN. This antibody was effective for Western blot detection of native BgGRN at concentrations of 1:5000.

### 2.5. Western Blot Analysis of BgGRN

Western blot detection of BgGRN was done using 250 ng protein suspended in Laemmli protein loading buffer. Samples were heated at 95 °C for 10 min, then loaded on 10% (vol/vol) SDS/PAGE gels and run on the Mini PROTEAN Tetra system (Bio-Rad, Hercules, California, USA) at 200 V and 180 mA. Samples were then blotted for 2.5 h onto 0.45 μm supported nitrocellulose membranes (Bio-Rad). Blocking was done for 1 h at room temperature in 5% (wt/vol) skimmed milk prepared in Tris-buffered saline (TBS) solution plus 0.1% Tween-20 (TBS-T buffer) before staining for 1 h in anti-V5 mouse IgG (recombinant BgGRN) or mouse anti-BgGRN IgG (native BgGRN) primary antibody at a concentration of 1:5000 in blocking buffer. Membranes were washed in TBS-T buffer for 10 min, then twice for 5 min each and once in TBS solution for 5 min. Membranes were then stained for 1 h in HRP-conjugated rabbit anti-mouse IgG antibody diluted 1:5000 in blocking buffer followed by a wash step as de- scribed earlier. Detection was accomplished by incubating the membranes in Super Signal West Dura Extended Duration substrate (Thermo Fisher Scientific). Chemiluminescent signals were acquired on the ImageQuant LAS 4000 machine (GE Healthcare).

### 2.6. Generation of Individual BgGRN Domains

Full-length rBgGRN (25 μg) was treated with 0.3 U/mL purified porcine elastase (Alfa Aeser) for a total of 18 h at 37 °C in order to assess whether the predicted elastase cleavage sites on BgGRN were accurate. A ponceau staining of the ruler and the solution was then completed to visualize the elastase solution ensuring cleavage. A 12 kda size exclusion column was then completed based on the predicted sizes of the individual domains in Table 2 to filter and purify the individual domains. The column flow through as well as the retained upper non-flow through was collected, and the cleavage products were shown on Western blot. Samples were run on SDS/PAGE and then blotted for 2.5 h onto 0.45-μm supported nitrocellulose membranes (Bio-Rad). Blocking was performed for 4 h at room temperature in 5% (wt/vol) BSA (Sigma, Oakville, ON, Canada) prepared in TBS-T buffer before probing with the anti-BgGRN antibody at 1:1000 dilution. Visualization of blots was accomplished as described earlier.

### 2.7. Cross Species Effect of BgGRN on S. mansoni Infection Success

In order to determine the cross-species activity of rBgGRN, aged-matched (shell approximately 8 mm in diameter) *Biomphalaria pfeifferi, sudanica*, and *glabrata* (M-line, BS-90, NMRI, and BB-02) were separated into two groups of 46 snails each per strain. Following the procedure outline by Pila et al. (2016),experimental groups received an injection through the shell into the haemoceol consisting of 100 nM full-length rBgGRN, while control snails received an identical amount of a non-cell proliferation inducing protein termed rBgTemptin, which was generated in an identical manner to rBgGRN [12,26]. Our 100 nM starting concentration was predicated on previous findings by Pila and associates (2016) [12], suggesting that 100 nM BgGRN concentrations are present during the early stages of infection, and are also capable of inducing resistance in M-line snails [12]. Openings caused by the needles used during injection were covered with wax. Once 48 hrs had passed after these treatments, snails were exposed to 5 of either PR-1 or NMRI *S. mansoni* miracidia over a 24-h period in individual wells of 12-well plates containing artificial spring water (ASW). The 48 hrs post treatment time was based on earlier work demonstrating that 48 h post granulin injection, a significantly higher amount of haemocytes are in circulation, and treatment of M-line snails with BgGRN 48 hrs prior to *S. mansoni* infection results in increased resistance levels [12].

Following a 24-h challenge, snails were transferred into tanks of ASW until 7 weeks post infection, a time at which differences in percentage of shedding snails after treatment with BgGRN has been shown to differ from that of control snails [12]. Snails were then examined after 12 h of exposure to light for the presence of shed cercariae.

### 2.8. Effects of BgGRN Subunits on Reactive Oxygen Species Generation

Haemocyte ROS production was measured in a manner similar to that of Humphries and Yoshino [2]. Haemocytes from BS-90 snails were isolated via the use of the head-foot retraction method. Haemolymph was pooled in 1.5 mL centrifuge tubes, on ice. Then, 100 uL aliquots were added the wells of black walled 96 well plate. Next, 100 uL of Chernin’s balanced saline solution (CBSS) was then added to each well. Cells were given 90 min at room temperature to attach to the bottom of the wells. Cells were then washed 4 times for 14 min with 200 uL of CBSS. Then, 100 uL of CBSS containing 100 nM of different granulin constructs (full length, intermediate domains, and individual granulin subunits) was added to the wells and allowed to incubate at RT for 2 h. Next, 100 nM of both Ionomycin and Phorbol 12-myristate 13-acetate (PMA) was utilized as a positive control. The 100 nM concentration was based off of previous work by Pila and associates (2016) [12]. Plates were centrifuged at 1500 rpm at 22 °C for 2 min, and supernatants were removed and place into a new black welled plate. H_2_O_2_ levels in the supernatant were then measured using an Amplex Red assay (Thermo Fisher Scientific, Mississauga, ON, Canada) as per the manufacturer’s instructions. Meanwhile, haemocytes were submitted to four half-hour freeze thaw cycles at −80 °C, after which their total double stranded (ds) DNA concentration was measured using a Quant-iT PicoGreen ds DNA Assay kit (Thermo Fisher Scientific, Mississauga, ON, Canada) as per the manufacturer’s directions. This allowed for H_2_O_2_ levels to be normalized to the total amount of dsDNA found in each well. Each experiment featured 3 replicates per treatment group. Experiments were run in triplicate.

### 2.9. Satistical Analysis

Differences between control groups and experimental groups in the cross-species infection data were measured via a standard Chi-squared test. Differences between treatment groups and the negative control in the ROS assay were assessed using multiple *t*-tests.

## 3. Results

### 3.1. Generation of Individual Subunits of BgGRN

Individual BgGRN domains were not expressed by Sf9 cells at any detectable amount, resulting in our usage of porcine elastase to generate the individual domains from full-length granulin. Full-length BgGRN was effectively cleaved in the presence of porcine elastase into a roughly 10 kda form. The cleaved form was selected via the use of a size exclusion column and demonstrated reactivity to our anti-BgGRN antibody (Figure 2).

### 3.2. Reactive Oxygen Species Production

BS-90 haemocytes produced H_2_O_2_ levels corresponding to 45.8 ± 11.8 RFU/ng of DNA in the absence of any stimulant. When primed with Ionomycin, and then induced to produce ROS with PMA, H_2_O_2_ levels jumped to 287.0 ± 38.8 RFU/ng DNA. The full-length granulin and the intermediates tested differed slightly from the levels of H_2_O_2_ seen in CBSS alone treatment, with the full-length granulin having the highest of H_2_O_2_ levels corresponding to 123.5 ± 58.1 RFU/ng DNA. The individual granulin domains generated via the use of porcine elastase and a size exclusion column generated levels of H_2_O_2_ that far exceeded any other recombinant, while also far surpassing hydrogen peroxide levels seen in the positive control. Cells treated with 100 nM of D1, D2, D3 and D4 produced 1270.5 ± 478.4 RFU/ng DNA, which varied significantly from the CBSS control (*p* < 0.0001) (Figure 3).

### 3.3. Cross Species Infection Reduction

The infection trials clearly demonstrated that BgGRN has the capacity to alter infection outcomes in a variety of *Biomphalaria* spp. and strains. Treatment with full-length BgGRN resulted in a significant (*p* < 0.05) reduction in infection prevalence in all four of these strains/species, regardless of the strain of *S. mansoni* used. NMRI *S. mansoni* failed to infect *B. sudanica*, whereas PR-1 *S. mansoni* infected 13% of control snails, and 8% of experimental snails. This 5% reduction was the only reduction observed that failed to be statistically significant (*p* = 0.214) Both NMRI and PR-1 *S. mansoni* failed to infect BS-90 snails. Differences in the baseline compatibility between the strains/species of *Biomphalaria* used in this study, with the two strains of *S. mansoni* used, were also observed. When challenged with NRMI *S. mansoni*, NMRI, M-line, BB-02 and *B. pfeifferi* snails were infected at higher rates in both treatment and control groups than when these same strains/species were infected with PR-1 *S. mansoni* (Figure 4).

## 4. Discussion

Granulins function as growth factors relevant in a variety of biological processes. In addition to their roles in inducing cell proliferation, they function in development, wound healing, tumorgenesis, and inflammation [11,13,15]. Our work suggests that in the *B. glabrata/S. mansoni* study model, fragments of the pro-BgGRN can serve as an immune cell activator, in addition to its previously described role as an inducer of cell proliferation. This study demonstrates that BgGRN impacts compatibility between *Biomphalaria* snails and *S. mansoni* in a multifaceted way, impacting the immune cell population as well as the ability of those cells to drive a ROS response.

Previous studies have demonstrated that pro-BgGRN induces the proliferation of adherent haemocytes that tend to express a specific toll-like receptor on their surface. In this study, we attempted to see if the proteolytically generated intermediate forms, as well as the single granulin subunit domains, would function in activating haemocytes. The results of our Amplex Red assay clearly demonstrate that BS-90 *B. glabrata* haemocytes respond to exposure with isolated single subunit BgGRN granulin domains by producing large quantities of H_2_O_2_. A similarly drastic increase in H_2_O_2_ production was not seen after the application of full-length pro-BgGRN, its three dual granulin domain-containing intermediate forms, nor the two three granulin domain-containing forms. The slight difference between these forms and the negative control may be due to cleavage of these recombinants into single subunits by the neutrophil elastase produced by the haemocytes in vitro. This finding remains consistent with previous reports detailing work done on proepithelins (PEPI) and their cleaved single granulin subunit forms known as epithelins (EPIs). In mice, PEPI inhibited Tumor Necrosis Factor (TNF) alpha-mediated activation of neutrophils, and prevented the production of oxidants and proteases, whereas EPIs induce the production of IL-8 by neutrophils [16]. IL-8 in turn possesses the capacity to potentiate reactive oxygen intermediate production [27,28].

Production of ROS has been shown to be a key function of circulating haemocytes within *B. glabrata*. It was previously demonstrated that several carbohydrates found on the surface of invading *S. mansoni* are capable of inducing ROS production by haemocytes, which in turn has resulted in *S. mansoni* evolving several ROS scavenging proteins [23,29]. Interestingly, haemocytes from susceptible *B. glabrata* have even been shown to be deficient in their capacity to generate ROS as compared to their resistant counterparts [30]. Haemocytes from *S. mansoni*-resistant BS-90 *B. glabrata* also increase the abundance of proteins necessary for ROS production when encapsulating sporocysts upon exposure to *S. mansoni* [31]. Our work suggests that in addition to a heightened capacity for generating ROS, resistant (BS-90) snails also generate single BgGRN domain fragments, which act as an endogenous signal capable of generating high levels of schistosome toxic H_2_O_2._ Such fragments do not appear to be present, or are at least present at lower concentrations, in M-line snails [12], suggesting that during the crucial 24–48 h period post infection by *S. mansoni*, M-line haemocytes are both reduced in the ability to generate parasite toxic H_2_O_2_ while also lacking BgGRN single subunits which act as potent inducers of ROS.

The mechanism by which these individual subunits enact their upregulation of ROS production remains unknown. Although granulins are known to activate MAPK signaling pathways, a definitive receptor for pro-BgGRN or BgGRN fragments remains unknown [11,12,15]. One possible set of candidates for a BgGRN receptor is *B. glabrata* TLRs. The *B. glabrata* genome is replete with leucine-rich repeat-containing molecules—at least 13 of which resemble canonical TLRs—while also containing numerous genes necessary for TLR mediated signaling [3,21,32]. One of these TLRs has been previously characterized as being an important haemocyte-associated component of the anti-*S. mansoni* immune response, and is also present at increase levels on haemocytes generated by injection of snails with full-length BgGRN [21]. TLRs are an interesting potential receptor for full-length BgGRN due to the discovery that mouse TLR9 uses granulin as a co-factor for binding CpG, although it remains to be seen whether this interaction is with a full-length granulin, or its individual subunits [33]. Regardless of the mechanism by which ROS generation is achieved, our work demonstrates that in addition to the hematopoietic and proliferative functions of pro-BgGRN, single subunit cleavage products of this growth factor function in activating haemocytes, thereby better preparing them to deal with invading sporocysts.

The structure of granulins across the animal kingdom is highly conserved. This high level of conservation exists as a result of the characteristic 12 cysteine repeat motif that acts as a signature for granulins [13]. The disulfide bonds between these cysteines are responsible for the tertiary structure within granulin domains, and their conserved nature means granulins retain a large amount of tertiary similarity, despite differences in primary amino acid sequences [18]. These similarities led us to hypothesize that the effects of BgGRN on the capacity of snails to fight of *S. mansoni* infections might be conserved across various strains of *B. glabrata* and various *Biomphalaria* species due to their evolutionary relatedness. We also had reason to believe that the recombinant granulin we generated from *B. glabrata* would retain its functionality across *Biomphalaria* species due to the fact that granulin from *Opisthorchis viverrini* is capable of inducing cell proliferation in humans, despite the distance between these two species [34,35].

Our work suggests that BgGRN retains the capacity to increase resistance to *S. mansoni* infection in all compatible combinations of *Biomphalaria* and *S. mansoni* examined. While BS-90 snails were completely refractory to infection with either PR-1 or NRMI *S. mansoni* independent of rBgGRN pre-treatment, the only other species/strain that did not feature a significant reduction in infected snails 7 w.p.i. was *B. sudanica.* The lack of a significant difference between granulin treatment and control treatment *B. sudanica* snails infected with PR-1 *S. mansoni* was likely due merely to the low infection rates observed, and not due to a lack of functionality by BgGRN. All other strains/species combinations yielded a decrease in shedding 7 w.p.i. of an average of 28.1% if the snail was pre-treated with BgGRN. These drastic decreases in infection prevalence were present for both PR-1 and NMRI strain *S. mansoni*, suggesting the mechanism by which resistance is elicited are not predicated on strain/strain compatibility, but by some alteration that is universally effective at combating *S. mansoni* infection. Difference in infection outcome may therefore be at least partially predicated on relative amounts of BgGRN production between species. Such an argument is supported by the finding that BS-90 BgGRN transcript levels increase significantly more when compared to M-line BgGRN transcript levels at one day post infection [12]. Additionally, this increase in resistance is not elicited by the insertion of irrational amounts of BgGRN, as our use of 100 nM rBgGRN was predicated on Pila and associates (2016) demonstrating that BS-90 snail plasma features 110.3 ± 16.3 nM BgGRN at day 2 post challenge [12].

BgGRN remains to date the only known molecular factor that has been shown to increase resistance to *S. mansoni* infection in a multitude of *Biomphalaria* species. This renders it an intriguing candidate for widespread use in rendering populations of intermediate snail hosts less likely to transmit the parasite to human populations. Highlighting the possibility of using the CRISPR Cas 9 system in order to overexpress BgGRN in wild snail populations by inserting a variant under the control of constitutive or infection induced promoters. The technology and methodologies necessary for such genetic modification exist, and are evidenced by the recent use of CRISPR in order to knock in and knock out genes in several snail species [36,37]. Giving wild populations of snails, which are typically highly resistant to infection in the first place, an additional tool to fight off schistosomes could prove to be an effective way to instituting a gene drive in order to control schistosomiasis. Gene drives have been shown to have some success in *Anopheles sp*., but only in laboratory settings to date [38,39,40]. Gene drives in snails are of additional interest due to the effects of schistosome infections on snails. Typically, snails are effectively castrated by such infections, which would theoretically be advantageous due to the increased likelihood of transmitting genes conferring resistance, such as the gene for BgGRN [8].

Worldwide, the amount of people infected with schistosomiasis has dropped from roughly ~259 million people in 2014 to ~206 million in 2016 [1]. Additionally, praziquantel treatment coverage for those infected has increased from ~20.7% to ~36.0% in the same time span. Despite these significant and promising advances in treating schistosomiasis, concern surrounding successful treatment and elimination remain due to the possibility of drug resistance emerging, as well as the high levels of re-infection seen post praziquantel treatment [41,42]. In order to properly address these concerns, and to further diversify the methods by which schistosomiasis is treated, we must further our understanding of the molecular determinants of host–parasite compatibility in the intermediate hosts of schistosomes. Our findings regarding the role that BgGRN has in eliciting ROS responses, as well as its capacity to increase resistance to infection in multiple *Biomphalaria* species renders it a molecule of key consideration moving forward in this domain.

## Figures and Tables

**Figure 1 genes-11-00038-f001:**
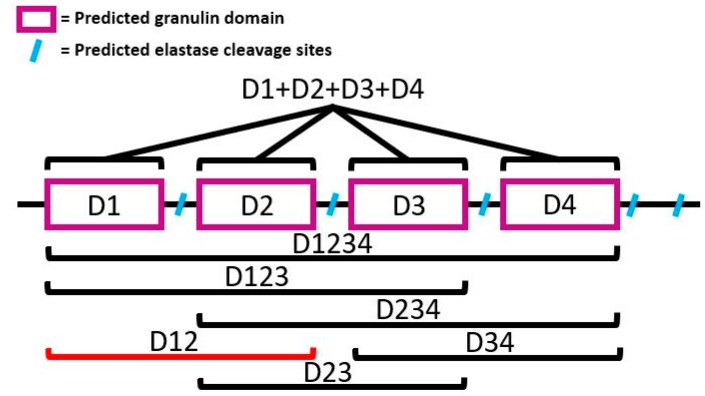
Diagram depicting the basic structure of *Biomphalaria glabrata* pro-granulin (BgGRN), the possible intermediate forms, as well as the fact that it can be cleaved into individual subunits. All generated granulin subunits feature a black bracket, whereas a red bracket indicates an intermediate that was not successfully cloned.

**Figure 2 genes-11-00038-f002:**
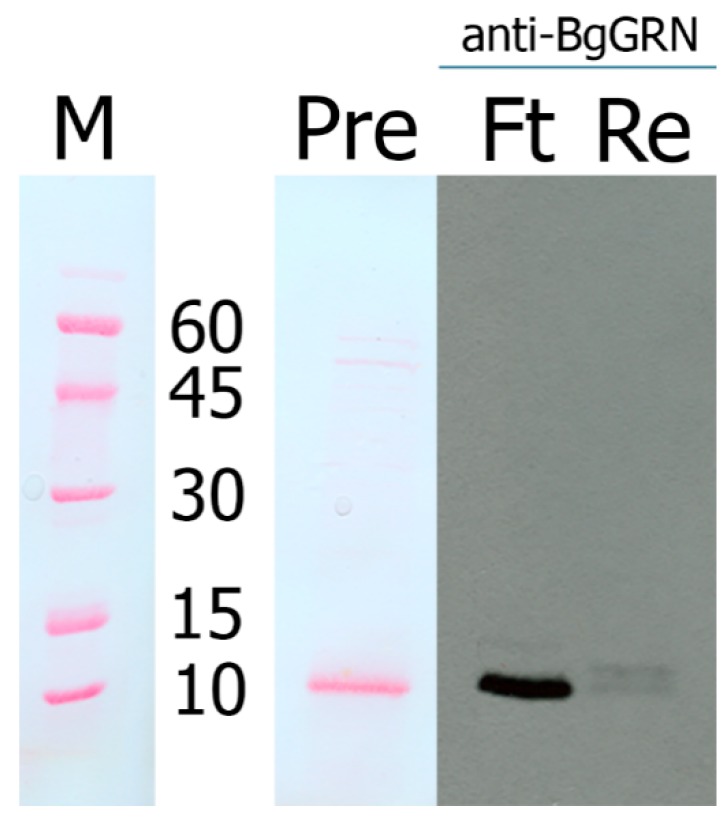
Ponceau stain of pre-column exclusion BgGRN treated with porcine elastase (Pre), alongside a Western blot using an anti-BgGRN antibody to demonstrate that individual granulin domains were successfully isolated in the column’s flow through (FT), and not present as much in the portion retained in the top of the column (RE).

**Figure 3 genes-11-00038-f003:**
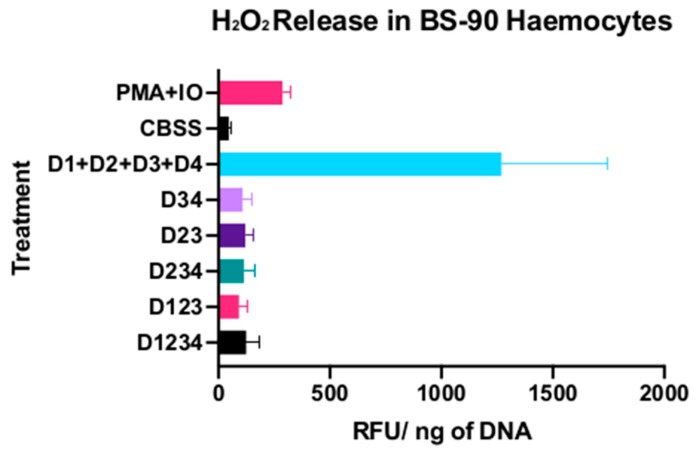
H_2_O_2_ production (*n* = 9) by BS-90 haemocytes in response to treatment with BgGRN (D1234), its intermediate forms (D34, D23, D123, D234), as well as its individual domains (D1,D2,D3,D4). Bars represent SD. Different letters indicate statistically significant (*p* < 0.05) differences between particular groups and different groups bearing other letters.

**Figure 4 genes-11-00038-f004:**
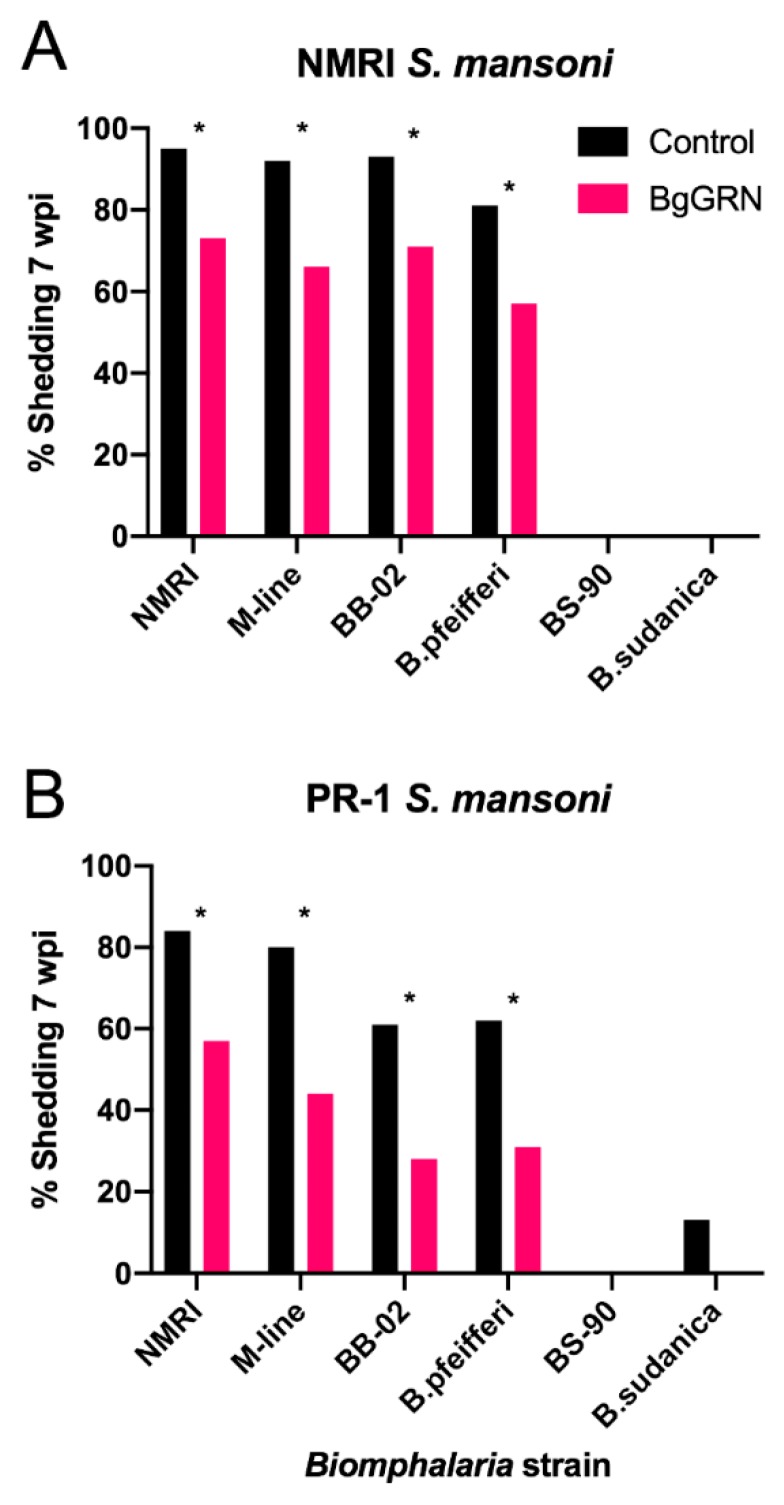
Infection prevalence seen across multiple strains of *B. glabrata*, as well as *B. pfeifferi* and *B. sudanica* in both control groups and groups having been injected with full-length BgGRN (*n* = 46 per treatment for each species/strain). Snails were infected with either (**A**) NMRI *S. mansoni* or (**B**) PR-1 *S. mansoni*. Significant differences (*p* < 0.05) between control and BgGRN treated groups as determined via a Chi-squared test are denoted using an asterisk.

**Table 1 genes-11-00038-t001:** List of primers used to generate full-length BgGRN, as well as its intermediate 2 and 3 granulin domain-containing intermediates. Red sequences indicate nucleotide additions necessary to directionally insert the PCR product into pENTR/D-TOPO, while blue sequences denote added ribosome-binding sequences.

BgGRN Domain1 + Full-Length Fwd2	C ACCAGG AGG GAC AAC TAC ATG ACT TGC TGC AAG GCT AAT
BgGRN Domain1 Rev2	GT AGC AAC GGC GGT TGA TCA
BgGRN Domain2 Fwd2	C ACCAGG AGG GAT GGA TCG ATG ACG TGC TGC CAG CTG GCT
BgGRN Domain2 Rev2	CT TCA CGC AGG TGC CAG CTG
BgGRN Domain3 Fwd2	C ACCAGG AGG GGT GGA GCT ATG ACT TGC TGC AAG CTC CAG
BgGRN Domain3 Rev2	CT TCT TGC ACT CGC CTT GAT
BgGRN Domain4 Fwd2	C ACCAGG AGG GAT GGT AAC ATG ACT TGC TGC AAG TTG GCC
BgGRN Domain4 + Full-Length Rev2	GC CCT TGT TGC ATG TTC CGG

**Table 2 genes-11-00038-t002:** Predicted sizes of full-length BgGRN, its 2 and 3 granulin domain-containing intermediates, as well as the individual granulin domains.

BgGRN Domains of Interest	Expected Molecular Weight
BgGRN Full length	44 Kda
BgGRN D1, D3, D4 separately	10 Kda
BgGRN D2	9.8 Kda
BgGRN D1D2	19.7 Kda
BgGRN D2D3	18.0 Kda
BgGRN D3D4	18.8 Kda
BgGRN D1D2D3	27.9 Kda
BgGRN D2D3D4	26.7 Kda

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
