# Peer review of "Biomphalaria glabrata Granulin Increases Resistance to Schistosoma mansoni Infection in Several Biomphalaria Species and Induces the Production of Reactive Oxygen Species by Haemocytes"

_genes, 2019, doi:10.3390/genes11010038_

Round 1
Reviewer 1 Report
The manuscript entitled 'Biomphalaria glabrata granulin increases resistance to Schistosoma mansoni infection in several Biomphalaria species and induced the production of reactive oxygen species by haemocytes' have well demonstration of how B. glabrata granulin (BgGRN) subunit(s) affect the S. mansoni development in intermediate host snail. They authors revealed the significant different of H2O2 induction from BS-90 haemocytes after BgGRN subunit(s) treatment. The authors have demonstrated that recombinant BgGRN would support snail immune response to decrease the S. mansoni infection and failure to develop/produce infective parasite stage to mammalian host by BgGRN injection into 3 different species of susceptible Biomphalaria sp. The knowledge of this work will be useful for the research community in the term of understanding about the host immune response to early stage of parasite infection. This study could be great information for further studies/develop tool for limiting/blocking the transmission of schistosome infection. This research works have well experimental design and sufficient results and need minor revision with comments as below.
Provide reference(s) for a sentence ‘A wide array of proteins have been identified…coordinate the immune response against mansoni’ (lines 49-50) Did sequence and annotation of BgGRN with predication of elastase cleavage sites publish? If so, please provide the reference. If not, I would suggest to move the figure into materials and methods section, before table 1. Please clarify how many recombinant BgGRN construct were made; five independent vector constructs (full length BgGRN, domain1, domain 2 domain 3 and domain 4) or four vector construct that were domains 1-4, but not full length BgGRN? And if full length BgGRN was constructed, what are the primers used. It is little confuse that at result section 3.1, there is indication that individual BgGRN domains were not expressed by Sf9 cells (line 195). Please clarify in figure legend of the abbreviation ‘Ft’ and ‘Re’ lanes in figure 2. Also, did non-digested full length rBgGRN would be probed by anti-BgGRN and revealing 44 kDa by western blot? Please include information of this point. Please clarify the results of section 3.2, if BS-90 haemocytes was treated with full length BgGRN or four domains 1-4. It seems like two similar sentences repeats on the lines 145-147 or rBgGRN was double digestion with two different vendor elastase? What is αBgGRN (line 155)? About section 2.8 Effects of BgGRN subunits on ROS generation, please consider to include the data or reference or discuss about H2O2 amount from BS-90 haemocytes after few days mansoni infection to compare the natural BgGRN function vs. recombinant ones. Please provide the statistic used for figure 4 in figure legends/text, numeric values of % shedding in result text, error bars in figures 4A and 4B. Please check tiny error typing, consistency, and abbreviation after full word use, for example, PMA (line 180) and hours or h (lines 136, 139 and 147)
Author Response
Reviewer # 1 comments
1.Provide reference(s) for a sentence ‘A wide array of proteins have been identified…coordinate the immune response against mansoni’ (lines 49-50).
In order to better cite the work being done in the field of snail/schistosome compatibility, we have included references encompassing two recent review papers published within the last 3 years in order to better represent novel discoveries in the field.
Pila EA, Li H, Hambrook JR, Wu X, Hanington PC. Schistosomiasis from a Snail’s Perspective: Advances in Snail Immunity. Vol. 33, Trends in Parasitology. 2017. p. 845–57.
Coustau C, Gourbal B, Duval D, Yoshino TP, Adema CM, Mitta G. Advances in gastropod immunity from the study of the interaction between the snail Biomphalaria glabrata and its parasites: A review of research progress over the last decade. Fish Shellfish Immunol. 2015 Sep 1;46(1):5–16.
2. Did sequence and annotation of BgGRN with predication of elastase cleavage sites publish? If so, please provide the reference. If not, I would suggest to move the figure into materials and methods section, before table 1.
The sequence had been previously published by Pila et al., who also predicted granulin subdomains, as well as neutrophil elastase cleavage points. To better reflect this, we have added a citation to their paper in this section, and have also mentioned their previous publication in the text. Additionally, we have altered Figure 1 so that it would serve as a clarification on what recombinants we were able to generate, while still highlighting the overall structure of BgGRN, as well as the intermediate and single subunit domains.
3.Please clarify how many recombinant BgGRN construct were made; five independent vector constructs (full length BgGRN, domain1, domain 2 domain 3 and domain 4) or four vector construct that were domains 1-4, but not full length BgGRN? And if full length BgGRN was constructed, what are the primers used. It is little confuse that at result section 3.1, there is indication that individual BgGRN domains were not expressed by Sf9 cells (line 195).
Five recombinant BgGRN constructs were made. One was the full-length construct, and the other four were intermediate forms containing either 2 or 3 of the granulin subunit domains. We have altered lines 105 and 106 to better reflect this point. Additionally, Table 1. Has been altered to indicate that the same Domain 1 Fwd primer was used for the full-length recombinant, while the Domain 4 Rev primer was also used for the full-length recombinant. In section 3.1, we describe the successful generation of each individual domain (D1, D2, D3,D4), unattached to each other, which we unfortunately could not get to work in Sf9 cells.
4.Please clarify in figure legend of the abbreviation ‘Ft’ and ‘Re’ lanes in figure 2.
Thank you for pointing out their absences. Lines 204 and 205 have been modified to include the abbreviations.
5.Also, did non-digested full length rBgGRN would be probed by anti-BgGRN and revealing 44 kDa by western blot? Please include information of this point.
Full length recombinant BgGRN as well as BgGRN from snail plasma were both recognized by our antibody in previous work done in our lab (Pila et al. 2016). Seeing as our antibody is already known to react with the full-length recombinant, we did not feel the need to include such a picture in this paper. Additionally, the lack of reactivity at ~44kda in the Western Blot portion of Figure 2 indicates that both our elastase digestion and our size exclusion column worked as predicted.
6.Please clarify the results of section 3.2, if BS-90 haemocytes was treated with full length BgGRN or four domains 1-4.
Whereas some cells were treated with the full length granulin, others were treated with intermediate forms, while others were treated with the individual subunits we generated using neutrophil elastase. We have added this information to the relevant materials and methods section (2.8) on line 181. The abbreviations used in Figure 3 have also been expanded upon in the figure legend in order to make this clearer.
7.It seems like two similar sentences repeats on the lines 145-147 or rBgGRN was double digestion with two different vendor elastase?
Thank you for pointing this out. The duplicate sentence was removed. Only one digestion with one type of elastase was used.
8.What is αBgGRN (line 155)?
In line 155, we are using the alpha symbol to denote “anti”, as in “anti-BgGRN”, as is standard designation for antibodies. To make this clearer, we have removed the alpha symbol and inserted “anti” in its place.
9.About section 2.8 Effects of BgGRN subunits on ROS generation, please consider to include the data or reference or discuss about H2O2 amount from BS-90 haemocytes after few days mansoni infection to compare the natural BgGRN function vs. recombinant ones.
We have expanded our discussion regarding ROS production by BS-90 snails in the discussion. It is difficult to compare absolute ROS values between our study and others that make use of the Amplex Red assay however, since the study conditions were not identical to those used in previous ROS-measuring studies.
10.Please provide the statistic used for figure 4 in figure legends/text, numeric values of % shedding in result text, error bars in figures 4A and 4B.
We included the statistical analysis used on figure 4 in our materials and methods section. In order to make this clearer, we have also altered Figure 4’s figure legend. Additionally, we have included the number of biological replicates in the figure legend as well. Seeing as this experiment was the result of one experimental trial with a relatively high number of biological replicates, standard error bars are unavailable.
11.Please check tiny error typing, consistency, and abbreviation after full word use, for example, PMA (line 180) and hours or h (lines 136, 139 and 147).
The requested alterations to improve consistency in abbreviations have been made.
Reviewer 2 Report
The manuscript “Biomphalaria glabrata granulin increases resistance to Schistosoma mansoni infection in several Biomphalaria species and induces the production of reactive oxygen species by haemocytes” by Hambrook et al. is a dualistic report that includes two separate parts. The first part is on the cloning, expression and production of full length granulin of B. glabrata and of its various shorter proteolytic cleavage products. This part is concluded with a preliminary functional testing of these proteins, which suggests that the mixture of the individual domains (D1, D2, D3, D4) is more active in inducing ROS production by haemocytes than each domain separately and even more active than the full length D1D2D3D4 protein. The second part of the study shows that the full length BgGRN can inhibit infection with S. mansoni in vivo. The results are clear, and encompass the testing of two strains of S. mansoni and six Biomphalaria strains with different susceptibility to S. mansoni. The results are however preliminary because only one dose of BgGRN is used, only one treatment schedule (injection in the haemocoel 48 h before infection), only one time after infection evaluated. Surprisingly, no experiments are run with the various shorter domains, including the highly active combination D1, D2, D3, D4. While it could be expected that the proteolytic cleavage would occur in vivo, the experiments with the individual domains and their mixtures would have helped clarifying the issue.
Minor points. The presentation of data is incomplete. In the Figures 3 and 4, data are possibly the mean of results obtained in replicate cultures/animals. This should be indicated, as well as the number of replicates. In Figure 3 the error bars are included but it is not said whether these are SD or SEM. Statistical significance is indicated with letters but what the letters mean is not declared. In Figure 4 the columns again probably represent means, and are without error bars. If the data are means of data from replicate animals should be stated, the error bars included, and the statistical significance indicated (adding asterisks without saying what asterisks mean is not sufficient).
Author Response
Reviewer #2
1.This part is concluded with a preliminary functional testing of these proteins, which suggests that the mixture of the individual domains (D1, D2, D3, D4) is more active in inducing ROS production by haemocytes than each domain separately and even more active than the full length D1D2D3D4 protein.
Unfortunately, we were unable to generate the individual domains separately, and were only capable of generating a mixture of all the single subunit domains using our elastase cleavage method. We were unable to test each domain separately, and were only able to test full length granulin, intermediate forms, and a mixture of the single subunits. The text (particularly the materials and methods) were altered in order to better reflect this point.
2.The second part of the study shows that the full length BgGRN can inhibit infection with S. mansoni in vivo. The results are clear, and encompass the testing of two strains of S. mansoni and six Biomphalaria strains with different susceptibility to S. mansoni. The results are however preliminary because only one dose of BgGRN is used, only one treatment schedule (injection in the haemocoel 48 h before infection), only one time after infection evaluated.
Thank you for raising this point. Although it is true that we only used 100nM concentrations of our BgGRN constructs in this paper, the decision to go with this concentration was informed by the results of previous work done in our lab. Pila et al. (2016) determined that, in BS-90 snails, BgGRN levels (as determined via ELISA) increase to 110.3 ± 16.3 nM at day 2 post challenge. This led us to determine that 100nM would be a realistic concentration to use when injecting snails for purposes of resistance experiments and would also be a biologically relevant concentration to use in our ROS assays. We have included discussion of this in lines 169-171. The decision to only use one treatment schedule was also informed by Pila et al.’s (2016) work, as it was determined that after 48 hours post BgGRN treatment, a significant increase in adherent haemocytes is seen. This treatment schedule is also consistent with that used in Pila et al infection outcome assessment between BS-90 and M-line snails. Finally, shedding was assessed at 7 weeks post infection seeing as that is the time at which Pila et al. (2016) saw shedding percentages plateau, thereby indicating that differences in shedding % were likely not to change after this point. We have included discussion of this in lines 174-177.
3.Surprisingly, no experiments are run with the various shorter domains, including the highly active combination D1, D2, D3, D4. While it could be expected that the proteolytic cleavage would occur in vivo, the experiments with the individual domains and their mixtures would have helped clarifying the issue.
Our investigation into the functionality of the individual granulin domains was based on previous observations in the field of granulin research suggesting that individual domains are functionally unique from their pro-granulin counterparts. Additionally, work done by Pila et al. (2016), which this paper is a continuation of, suggests that individual granulin domains are indeed generated in some snails (particularity in BS-90 snails. Unfortunately, as mentioned in our materials and methods, we were unable to generate the individual granulin subunits via traditional cloning methods, and therefore could only generate a mixture of the single granulin domains using our elastase cleavage procedure.
4.The presentation of data is incomplete. In the Figures 3 and 4, data are possibly the mean of results obtained in replicate cultures/animals. This should be indicated, as well as the number of replicates. In Figure 3 the error bars are included but it is not said whether these are SD or SEM. Statistical significance is indicated with letters but what the letters mean is not declared.
We have reworked the legend of Figure 3 to include all of the requested information. In addition to this, we have added a few sentences to the materials and methods section regarding our ROS assay so that the number of replicates per trial as well as the number of experimental replicates would be clearer.
5.In Figure 4 the columns again probably represent means, and are without error bars. If the data are means of data from replicate animals should be stated, the error bars included, and the statistical significance indicated (adding asterisks without saying what asterisks mean is not sufficient).
Due to the precedent set by Pila et al. (2016) and the fact that this study required coordination between numerous snail and S. mansoni strain/species in order to challenge an age-matched cohort of snails of each group, we decided that our cross-species resistance examination would be done with a single experimental replicate including a relatively high number of biological replicates (many individual snails in each group). For these reasons, the figure reflecting this study lacks error bars. We have also amended the figure legend to include the number of biological replicates used in this experiment. To your point regarding asterisk, we acknowledge that a failure to indicate what they stood for was an oversight on our part, and so we have adjusted the figure legend accordingly. We have also included the description of the Chi-squared test used in analyzing our cross-species resistance data in the Materials and Methods section.
Reviewer 3 Report
General comments: In this manuscript, Jacob R. Hambrook et al. reported that BgGRN enhances host defense against Schistosoma mansoni infection in Biomphalaria by the stimulation of haemocytes to produce reactive oxygen species. They showed that BS-90 haemocytes can induce H2O2 production in response to treatment with BgGRN. This study suggests that BgGRN may serve as an immune activator in host defense. This Ms. Is well organized and written, I recommend its publication after minor revision. Specific comments: 1. In Fig.4, how many biological replicates are performed should be indicated in the legend and the error bars should be given. 2. The authors should detect the expression level (mRNA or protein) of endogenous BgGRN in real Schistosoma mansoni infection of Biomphalaria glabrata, which may further support their conclusion that BgGRN is an immune cell activator.
Author Response
Reviewer #3
1. In Fig.4, how many biological replicates are performed should be indicated in the legend and the error bars should be given.
Thank you for pointing this out. We have taken your advice and have inserted the number of biological replicates into the figure legend for figure 4. Seeing as we decided to go with a single experimental replicate with a relatively high number of biological replicates, there are no error bars to report.
2. The authors should detect the expression level (mRNA or protein) of endogenous BgGRN in real Schistosoma mansoni infection of Biomphalaria glabrata, which may further support their conclusion that BgGRN is an immune cell activator.
This is a great point, and we are fortunate that previous work in our lab (Pila et al. 2016) has demonstrated that in both susceptible (M-line) and resistant (BS-90) strains of B. glabrata, BgGRN transcript levels increase sharply within 48 hours of infection with S. mansoni. This is significant as it demonstrates, as you suggest, that BgGRN is a likely immune cell activator. Interestingly BS-90 snails have a higher transcript abundance that M-line snails, implicating BgGRN’s role in resistance even further. In order to make the necessary connection between Pila et al’s findings and our own, we have included a few sentences addressing this point, starting at line 313.
Round 2
Reviewer 2 Report
no further comments